# Phage and Antibiotic Combinations Reduce *Staphylococcus aureus* in Static and Dynamic Biofilms Grown on an Implant Material

**DOI:** 10.3390/v15020460

**Published:** 2023-02-07

**Authors:** Hyonoo Joo, Sijia M. Wu, Isha Soni, Caroline Wang-Crocker, Tyson Matern, James Peter Beck, Catherine Loc-Carrillo

**Affiliations:** 1Micro-Phage Laboratory, Department of Internal Medicine, University of Utah, Salt Lake City, UT 84132, USA; 2Department of Veterans Affairs, Salt Lake City Health Care System, Salt Lake City, UT 84148, USA; 3Department of Orthopaedics, University of Utah, Salt Lake City, UT 84108, USA

**Keywords:** bacteriophage, biofilm, vancomycin, combination therapy, *Staphylococcus aureus*

## Abstract

*Staphylococcus aureus* causes the majority of implant-related infections. These infections present as biofilms, in which bacteria adhere to the surface of foreign materials and form robust communities that are resilient to the human immune system and antibiotic drugs. The heavy use of broad-spectrum antibiotics against these pathogens disturbs the host’s microbiome and contributes to the growing problem of antibiotic-resistant infections. The use of bacteriophages as antibacterial agents is a potential alternative therapy. In this study, bioluminescent strains of S. aureus were grown to form 48-h biofilms on polyether ether ketone (PEEK), a material used to manufacture orthopaedic implants, in either static or dynamic growth conditions. Biofilms were treated with vancomycin, staphylococcal phage, or a combination of the two. We showed that vancomycin and staph phages were able to independently reduce the total bacterial load. Most phage-antibiotic combinations produced greater log reductions in surviving bacteria compared to single-agent treatments, suggesting antimicrobial synergism. In addition to demonstrating the efficacy of combining vancomycin and staph phage, our results demonstrate the importance of growth conditions in phage-antibiotic combination studies. Dynamic biofilms were found to have a substantial impact on apparent treatment efficacy, as they were more resilient to combination treatments than static biofilms.

## 1. Introduction

A number of medically important bacteria have the ability to create a protective layer around themselves, forming biofilms (bacterial communities) on foreign materials that can evade the host’s immune system. In the United States alone, an estimated 14 million biofilm-related infections lead to over 350,000 deaths annually [1]. Furthermore, many biofilm-causing bacteria are antibiotic-resistant and lead to medical implant failures [2]. Thus, there is an increasing need for alternative strategies to combat biofilm-associated infections. Although there have been improvements in prophylaxis and sterile surgical techniques, infection continues to be a costly and challenging problem that plagues orthopaedic implant surgery [3].

Staphylococci are the predominant bacteria that cause implant-related infections in orthopaedic surgery [4,5,6]. The upregulation of specific genes in *S. aureus* enables bacteria to adhere to implant surfaces, aggregate, and form an extracellular polymeric substance (EPS) matrix [7,8]. Biofilm adherence to biomaterials is mediated by surface protein adhesins (i.e., microbial surface components recognizing adhesive matrix molecules—MSRAMM) [9,10], and biofilms are formed when bacterial cells form multiple layers over the substrate surface via intercellular adhesion. After adherence, dispersion of biofilms allows bacterial cells to spread to uncolonized sites within a host [11].

Biofilm infections are particularly difficult to treat because biofilms can better withstand the host’s immune responses compared to planktonic cells [12,13]. An example of this was seen in a mouse model used to study catheter-associated infections—staphylococcal biofilms were able to re-program the host immune response from a typical pro-inflammatory response to an anti-inflammatory, pro-fibrotic response which allowed biofilms to persist in vivo [14,15]. Other studies have shown that biofilm infections are diverse, resulting in unique immune responses to staphylococcal biofilms based on the infection site’s tissue microenvironment [15]. In addition, the use of antibiotics to treat chronic infections has been found to be less effective against biofilm bacteria [16,17,18]. Biofilm-conferred antibiotic resistance mechanisms include: (1) reduced antibiotic penetration through the biofilm matrix, (2) differentiation of certain biofilm cells into a protected phenotype, (3) upregulation of antibiotic-efflux pumps, and (4) antibiotic inactivation from beta-lactamase enzymes [19,20].

The use of surgical implants has drastically increased due to technological advancements and their ability to significantly improve quality of life. For example, hip and knee replacements are among the most common and fastest growing surgical procedures in U.S. hospitals [21]. Due to an aging population, the high demand for orthopaedic devices will likely continue to increase. A conservative analysis shows that the number of total joint replacements will double between 2014 and 2030 [22]. Despite their therapeutic benefits, implant surfaces are foreign biomaterials that can be colonized by bacteria in the form of biofilms. To the detriment of patients, these implant-related biofilm infections can necessitate invasive surgery and prolonged use of antibiotics [23,24].

First implemented as a treatment against bacterial infections in the early 20th century, phages were largely dismissed in favor of chemical antibiotics. Due to the rise of antibiotic-resistance, however, they are being reconsidered as potent therapeutic agents. Phages are self-reproducing viruses that target bacteria much more selectively than antibiotics [25,26]. Phages have been viewed favorably because of their lack of cross-resistance with antibiotics, host range specificity, and minimal disruption to the human microbiome [25]. They are also environmentally ubiquitous and easily isolated from soil or sewage samples [27,28,29]. Phage therapy has been tested in select clinical trials and used in rare cases under compassionate care exceptions [30]. Limited to a small number of patients, some early studies report total bacterial eradication, while others demonstrate little to no therapeutic effect [31]. These seemingly contradictory patient outcomes suggest the need for further research. Because phage therapy will likely be used in conjunction with antibiotics, there is a natural and practical interest in studying the interactions between phages and antibiotics. 

There has been increasing interest in studying the efficacy of combining antibiotics with phages against biofilms—readers are referred to reviews by Tagliaferri et al. and Abedon on the topic [32,33]. However, most of these studies use static growth conditions to grow biofilms on materials not found in orthopaedic implants. To test treatments in more accurate in vitro models, we used dynamic growth conditions and a clinically relevant biomaterial: poly-ether-ether-ketone (PEEK). Of clinical interest, PEEK is a synthetic polymer that displays high temperature resistance, biocompatibility, radiolucency and wear resistance [34,35]. Compared to metal implants, PEEK has an elastic modulus closer to that of bone. This avoids the stress-shielding effect, where stiffer metal biomaterials cause implant-adjacent bone resorption which leads to device failure [35,36]. These properties make PEEK a desirable material for making prosthetic devices such as spinal implants [37].

To address gaps in prior research, our goal was to determine if a phage-antibiotic combination treatment would have a synergistic, additive, or antagonistic effect on *S. aureus* biofilms grown on PEEK materials. Furthermore, we investigated the impact that growth conditions would have on anti-biofilm efficacy. Vancomycin alone, staph phage K alone, and vancomycin-staph phage K combination treatments were applied to static and dynamic PEEK membrane biofilms. By using a bioluminescent strain of *S. aureus* (i.e., Xen 29), we were able to simultaneously assess both metabolic activity and viable bacterial counts. Furthermore, two novel phages were tested in combination with vancomycin against MRSA dynamic biofilms grown on PEEK discs.

## 2. Materials and Methods

### 2.1. Microorganisms, Culture Conditions, and Antibiotics

*Staphylococcus aureus* subsp. *aureus* Xen 29 (Caliper LifeSciences, Alameda, CA, USA) was used as the biofilm-forming organism for the PEEK membrane experiments in this study. Its parent strain, *S. aureus* (ATCC 12600), was modified to contain a *Photorhabdus luminescence luxABCDE* operon at a single integration site in order to luminesce when metabolically active, and re-named *S. aureus* Xen 29 [38]. For the PEEK disc coupon biofilms, we used methicillin resistant *Staphylococcus aureus* (MRSA) strain SAP231 (obtained from Roger Plaut, FDA). This strain was created by transducing NRS384 with a modified lux operon [39]. Both stocks were maintained at −80 °C and propagated on LB plates (Luria-Bertani, Miller broth mixed with 1.2% (*w/v*) Technical Agar; Difco, Sparks, MD, USA), and incubated at 37 °C overnight prior to use for experiments.

*Staphylococcus aureus* bacteriophage K (ATCC^®^ 19685-B1^TM^; aka Staph phage K) was used on the MSSA membrane biofilms. Staph phages WTP113011 and WTP092811 were used on the MRSA disc biofilms. These ‘WTP’ phages were isolated in-house from Salt Lake City Water Reclamation Plant, using the single plaque purification method and the *S. aureus* strain SAP231 as host.

All three phages were propagated using standard amplification methods to achieve a titer of approximately 1 × 10^10^ PFU/mL. Briefly, an overnight culture of *S. aureus* Xen 29 was added 1:100 in volume to Brain Heart Infusion (BHI; Difco, Sparks, MD, USA) broth and grown for approximately 2–3 h until an optical density (OD_600_) of 0.2 was obtained, which was confirmed to be ~1 × 10^8^ CFU/mL using a 10-fold serial dilution technique. Diluting the inoculum by no more than 20% (*v/v*), staph phage K was added such that the phage to bacteria ratios (i.e., multiplicity of infection (MOI)), was between 0.01 and 0.10. Cultures were incubated at 37 °C with shaking at 200 rpm for 2 h and then allowed to stand at room temperature (~25 °C) overnight to slow bacterial growth. The following day, the remaining bacteria were pelleted by centrifugation at 2000× *g* for 30 min. The supernatant containing the phages was filtered using a membrane filter (Millipore, Billerica, MA; 0.22 µm) and stored at 4 °C.

To determine phage titers, 100 µL samples were mixed with 200 µL of a *S. aureus* Xen 29 overnight culture (~2 × 10^9^ CFU/mL), and this mixture added to molten soft agar (Luria-Bertani, Miller broth mixed with 0.6% (*w/v*) Technical Agar) tempered at 55 °C and overlaid on an LB plate. Once the soft-agar had set, plates were incubated overnight at 37 °C. The following day, plaques were enumerated using the 10-fold serial dilution techniques to determine the number of plaque forming units (PFUs).

Vancomycin hydrochloride from *Streptomyces orientalis* (Sigma-Aldrich; St Louis, MO, USA) was made into a concentration of 1 mg/mL using deionized water, then filter-sterilized (0.22 µm), and stored at 4 °C. For the treatment of biofilms, vancomycin was added to biofilm-maintaining media (i.e., BHI), to obtain various final concentrations ranging from 9 µg/mL to 42 µg/mL. These concentrations were reported to be the average trough and peak concentrations of vancomycin present in rat serum during a pharmacokinetics study mimicking human intravenous administration of 1 g of vancomycin every 12 h [40], attempting to achieve ~24 mg/kg per day [41].

### 2.2. In-Vitro Biofilm Models on PEEK Membrane

In this study, two different in-vitro models were used to compare the effects of the treatments on biofilms grown on PEEK membranes. For the first model, dynamic biofilms were grown on PEEK material using a CDC biofilm reactor [42] (Figure 1A). For the second model, static biofilms were grown on PEEK material in microtiter plate wells (Figure 1B). Poly-ether-ether-ketone (PEEK) was selected for the material to grow biofilms on, as previous studies have demonstrated its biocompatibility and suitability for use in orthopaedic implants [37,43,44]. PEEK woven membrane sheets contained 200 µm thread diameter spaced 300 µm apart (Small Parts™—discontinued).

PEEK material (12 mm × 12 mm squares) was cut from the membrane sheets and fitted in a modified CDC biofilm reactor rod (Bio-surface Technologies, Bozeman, MT, USA). Most other bioreactor components were unmodified. The modified rods were designed and constructed for holding the PEEK material in place. The cylindrical bases of each rod were tapered with wedges of approximately 5 mm (W) × 4 mm (H), making for an easier fit through the slotted lid. The inward face of the 55 mm distal end was designed with three square 100 mm^2^ openings. The outward face was fitted with grooves to allow the PEEK material to be secured between two stainless steel components: three 15 × 15 × 1 mm squares with 100 mm^2^ openings and one 55 × 18 × 1 mm plate with three 100 mm^2^ openings, including a screw-fitting hole used to secure the metal components to the cylindrical base of the rod (Figure 1B).

*S. aureus* Xen 29 was subcultured from a −80 °C stock onto LB plates and incubated overnight at 37 °C. The following day, 2–3 colonies were transferred and suspended in BHI broth to produce a turbid suspension with an OD_600_ of ~0.2. The fully-assembled biofilm reactor (Figure 1A) was filled with 500 mL of BHI and sterilized. The 1 mL liquid culture of Xen 29 was used to inoculate the reactor, which was placed on a digital stirring hot-plate (Super-Nuova; Thermo Scientific, Waltham, MA, USA) to maintain a constant surface temperature of 34 °C (uniform broth temperature of 28.5 °C), stirring at 130 rpm, for 48 h to form dynamic biofilms.

For static biofilms, *S. aureus* Xen 29 was subcultured in the same fashion as before. The following day, 2–3 colonies were suspended in 3 mL BHI broth and incubated overnight at 37 °C with moderate shaking (200 rpm). The liquid culture was then diluted 1:100 using BHI broth, and 5 mL of the inoculate was dispensed into individual wells, containing a piece of sterilized PEEK membrane (Figure 2). Multi-well plates were then incubated at 37 °C for 48 h to form static biofilms.

### 2.3. Antimicrobial Treatments of Biofilms

Once biofilms were formed on the PEEK membrane, planktonic cells were washed off the membranes using gentle agitation in phosphate buffered saline (PBS; MP Biomaterials, Santa Ana, CA, USA). Membranes were then transferred to individual culture plates (35 × 10 mm) and maintained in 2 mL BHI broth.

Biofilms were treated with antimicrobials introduced to the maintenance media. The treatments included: vancomycin (42 µg/mL); staph phage K (~3.5 × 10^9^ PFU/mL); or a combination of antibiotic and phage at varying concentrations as presented in the results below. In general, the addition of antimicrobial treatments to BHI did not exceed 20% (*v/v*) of the media.

Over a 72-h treatment period, culture plates containing biofilms were maintained at room temperature (approximately 25 °C), and the media was replenished at 24 and 68 h to minimize evaporation and keep the biofilms hydrated.

### 2.4. Bioluminescent Imaging

After treatment, biofilms were visualized for bioluminescent activity, a good indicator of the metabolic activity of the cells [22], using the in vivo imaging system (Lumina II, Caliper Life Sciences, Alameda, CA, USA) with the following parameters: binning 8, f-stop 2.5, and 3-min exposure.

### 2.5. Quantification of Viable Counts

Following imaging, the number of viable bacteria and phages on the PEEK membranes, as well as in the maintenance media, were enumerated. PEEK membranes were removed from maintenance media, and planktonic cells as well as non-biofilm-adherent phages were washed 3× by gentle agitation of the membranes in PBS. After washing, each membrane was transferred to a 10 mL glass tube containing 2 mL of PBS. These were vortexed for 30 s and then sonicated for 10 min using an ultrasonic water bath (Bransonic 42; Branson, Danbury, CT, USA). The sonicated bacteria were allowed to rest for 5 min before supernatant was ten-fold serially diluted using PBS. Serial dilutions were seeded onto LB plates and incubated overnight at 37 °C. Phage counts were enumerated in a similar way using ten-fold serial dilution and the soft agar overlay method. Vortexing the samples for 30 s was found to be statistically insignificant (*p* > 0.95) at reducing phage counts for staph phage K. The following day, colony forming units (CFU) and plaque forming units (PFU) were counted to determine the total number of bacteria and phages present on the membranes and in the maintenance media.

### 2.6. Treatment and Processing Biofilms on PEEK Disc Coupons

To further test the surface properties of antibiotic and phage combination treatments on PEEK materials, we used SAP231 to grow MRSA biofilms. The biofilms were grown dynamically in a similar fashion as described above. However, the biofilms were developed on solid PEEK disc coupons (Bio Technologies, Bozeman, MT, USA), measuring 12.7 mm in diameter and 3.8 mm in thickness, instead of PEEK membrane. After 48 h of incubation in the CDC bioreactor, the coupons were washed in PBS and transferred to a 24-well plate (CELLTREAT Scientific Products, Pepperell, MA, USA) that housed the treatment-media mixtures in LB with 1% glucose (*w/v*). Over a 72-h treatment period, these biofilms were treated with staph phages WTP113011 and WTP092811 (10^8^ PFU/mL), and vancomycin (40 µg/mL).

After 72 h, the coupons were washed in PBS and then transferred to sterile glass tubes containing 2 mL of PBS. The tubes were vortexed for 1 min and sonicated for 15 min. The sonicated bacteria were allowed to rest for 20 min before the suspension was vortexed for another minute. The supernatant was serially diluted and seeded onto LB plates that were incubated overnight at 37 °C. Colonies were counted the following day to calculate the CFU per disc coupon.

### 2.7. Statistical Analysis

Statistical analyses were performed on log_10_ transformations of CFU and PFU counts. Statistically significant differences between treated and untreated groups as well as between various treatment groups were evaluated using a Student’s unpaired two-tailed *t*-test, assuming equal variance with an alpha level of 0.05.

Changes in the bacterial (CFU) counts due to the combinatorial treatments were categorized as either synergistic, additive, or antagonistic. Synergistic combinations demonstrated greater bacterial reduction than the sum of the individual treatments. Additive combinations demonstrated approximately equal bacterial reduction compared to the sum of the individual treatments. Antagonistic treatments resulted in smaller bacterial reduction compared to the sum of individual treatments. In order to statistically evaluate these effects, the following equation was used [45]:
$$C\;=\;log\;(AP^{R})\;-\;(log\;(A^{R})\;+\;log\;(P^{R})),$$
where

C: coefficient of interaction

A^R^: reduction in bacteria counts antibiotic treatment

P^R^: reduction in bacteria counts for phage treatment

AP^R^: reduction in bacterial counts following the combined treatment (AP).

The coefficient of interaction, C, indicates synergy or antagonism:

C = 0: additive interaction (1)

C > 0: Synergistic interaction (2)

C < 0: Antagonistic interaction (3)

A linear regression analysis was performed to determine the significance of phage and antibiotic interactions using the Python *statsmodels* library [46].

## 3. Results

### 3.1. Static versus Dynamic Biofilms

The bacterial loads of biofilms grown under static versus dynamic conditions were enumerated following an initial 48-h growth period (i.e., t = 1 h). Dynamic growth conditions consistently produced biofilms with higher bacterial loads compared to static conditions, with an average total bacterial load of 1.68 log CFU greater for dynamic biofilms than that of the static biofilms (*p* < 0.001). However, at 72 h after the initial growth period, the difference narrowed to 0.29 log CFU (*p* = 0.003) (Figure 1).

When biofilms were treated with staph phage K at 10^9^ PFU/mL, the multiplicity of infections (MOIs) were found to be 0.02 and 0.004 for the static and dynamic biofilms, respectively. The average phage titer was 1.08 log PFU/mL greater in the dynamic biofilms compared to the static biofilms at t = 1 h (*p* < 0.001). However, the average concentration of staph phage K was 0.29 log units greater in the static biofilms compared to the dynamic biofilms at t = 72 h (Figure 2).

*S. aureus* biofilms were treated with staph phage K at 10^9^ PFU/mL, or vancomycin 42 µg/mL, or a combination of staph phage K and vancomycin. Compared to the control groups, all treatment types significantly decreased bacterial load in the biofilms, with a greater decrease observed in the static growth set. Compared to the untreated control group, treatment with staph phage K, vancomycin, and combined staph phage K with vancomycin decreased the bacterial loads in dynamic biofilms by 0.4, 1.3, and 3.2 log CFU, respectively. In addition, the largest difference between the static and dynamic biofilms (1.7 log CFU) was observed in the combinatorial treatment condition (Figure 3).

The combined treatment synergistically decreased bacterial concentration in both the static and dynamic biofilm conditions (*p* = 0.008, *p* < 0.001).

### 3.2. Combination Treatment of Dynamic Biofilms

Compared to the untreated control group, treatment with vancomycin 42 µg/mL + staph phage K at 10^7^, 10^8^, or 10^9^ PFU/mL decreased bacterial loads by 0.80, 1.22, and 3.20 log units, respectively. All treatments resulted in statistically unique reductions of *S. aureus* CFU except for the treatments containing 10^7^ and 10^8^ PFU/mL staph phage K. 

Compared to the average of the untreated control group, treatment with staph phage K at 10^9^ PFU/mL and vancomycin at 9 µg/mL, 17.25 µg/mL, 33.75 µg/mL and 42 µg/mL reduced the bacterial loads by 0.49, 0.39, 1.58, and 3.20 log units, respectively. Similar to the varied phage concentrations, all treatments resulted in statistically distinct reductions in bacteria except between 9 and 17.25 µg/mL vancomycin (Figure 4).

Overall, the largest bacterial decrease occurred when using a phage titer of 10^9 PFU/mL and a vancomycin concentration of 42 µg/mL.

### 3.3. Metabolic Activity of Biofilms over 72 h

The bioluminescent *S. aureus* Xen 29 strain used, allowed us to monitor the metabolic activity of its biofilms over the 72-h treatment period. Both vancomycin only and combinatorial treatments were able to reduce the metabolic activity below the imaging system’s limit of detection. The staph phage K treated petri dishes only show bioluminescence around the mesh, whereas the untreated petri dishes show the bacteria dispersed from the mesh into the media as planktonic cells. Interestingly, the metabolic activity appeared to fluctuate after the growth media was replenished (t = 24 h) during treatment (Figure 5).

### 3.4. Combination Treatment of Biofilms on PEEK Disc Coupon

Vancomycin significantly reduced the bacterial load on the disc biofilms compared to the untreated group. However, when applied by themselves, staph phages WTP113011 and WTP092811 failed to significantly reduce MRSA in the disc biofilms. Although the two phages by themselves killed bacteria very comparably, they showed distinct patterns of interaction when combined with vancomycin at 40 µg/mL. Phage WTP113011 did not demonstrate significant synergy, while phage WTP092811 showed significant synergy when combined with vancomycin. In several trials, phage WTP092811 and vancomycin completely prevented any detectable MRSA growth, indicating that the combination reduced the bacterial load below our limit of detection, i.e., 2 log CFU (Figure 6).

## 4. Discussion

To our knowledge, our study is the first of its kind to directly compare the efficacy of phage and antibiotic combinations on dynamic and static biofilms. The CDC bioreactor allowed us to compare PEEK membrane biofilms grown in a dynamic, turbulent environment compared to those grown in a static culture. Immediately after applying the combination treatments to both types of biofilms, we recovered a higher phage titer from dynamic biofilms than static biofilms even though the initial treatment titer was equivalent (~10^9^ PFU/mL). This disparity in phage uptake may lie in architectural differences between the two types of biofilms. The turbulent flow that dynamic biofilms were grown in can affect biofilm structure, causing them to be denser and more strongly attached to their host surface than static biofilms [47,48,49]. This distinct architecture, which can include water channels, may help phages more freely infiltrate dynamic biofilms. Despite this initial difference, static and dynamic biofilms had similar phage titers by the end of the 72-h treatment period.

In this study, the bacterial load of static biofilms was significantly smaller than that of their dynamic counterparts for all treatment conditions. The largest disparity between static and dynamic results was observed during the combined treatment of phage and antibiotics. These results suggest that conducting experiments with static biofilms could lead to a false sense of biofilm-eradicating efficacy, especially in combinatorial therapies. Future phage and antibiotic combination studies should consider the effect that dynamic growth conditions can have on observed treatment efficacy. 

When compared to static biofilms, however, dynamic biofilms require additional materials, equipment, and time. For example, as little as 50 µL of growth media is required to grow a biofilm using traditional static methods [45]. Conversely, dynamic biofilms grown in this study required a steady supply of fresh media, consisting of at least 1250 mL of BHI per bioreactor. Factors such as smaller sample sizes and longer growth periods may present significant barriers to quickly testing a wide array of treatments against biofilms. Further research is needed to assess the viability of using static, instead of dynamic, biofilms to facilitate high throughput treatment screening.

Device material can significantly impact the development of biofilm infections [50]. In many experiments, however, biofilms are grown on polystyrene plastic instead of biomaterials used in implantable medical devices [51,52,53,54]. In order to simulate a pathologically relevant surface-environment, we conducted experiments using PEEK, a biomaterial used in orthopaedic implants [37,55,56]. We later used solid PEEK disc coupons because they more closely resemble the surface geometry of PEEK used in medical devices. In the disc coupon experiments, we also used MRSA—a serious threat in antibiotic resistance according to the Center for Disease Control and Prevention [57]—as the biofilm-forming organism instead of MSSA, as vancomycin is usually reserved to treat MRSA infections in clinical applications [58].

Using PEEK disc coupons, we tested two newly isolated staph phages with and without vancomycin. We found that only one of the two phages (WTP092811) demonstrated significant synergy with vancomycin. The results from the PEEK disc coupon experiments reflect the importance of phage selectivity in combination treatments. While single-agent treatment results of both phages were comparable, phage WTP092811 combined with vancomycin resulted in a larger *S. aureus* reduction than phage WTP113011 combined with vancomycin. Other studies have shown that antibiotics and phages can demonstrate antagonism, where the combination treatment resulted in less effective bacterial reductions than the sum of the individual treatments [32,51,52]. Taken together, these findings indicate that phage and antibiotic identities greatly affect the nature and efficacy of combination treatments.

There are many potential benefits to combining phage and antibiotic therapies, including lowering the effective concentrations required of antibiotics and suppressing antibiotic resistance [59]. In 2007, Comeau et al. published a study highlighting phage and antibiotic synergy. They demonstrated that sublethal concentrations of certain antibiotics increased the progeny and burst size of MFP phages and T4-like phages against *E. coli* bacteria [60], this PAS (phage antibiotic synergy) effect could be considered advantageous in biomedical applications where increasing the number of phage produced could result in higher phage dosing in conjunction with the administration of subtherapeutic antibiotic dosages. In our first set of experiments, we tested staph phage K and vancomycin—independently and simultaneously—against static and dynamic *S. aureus* biofilms grown on PEEK membrane. Based on the concentrations tested on membrane-based biofilms, we found that there was a correlation between bactericidal activity and the concentrations of both staph phage K and vancomycin. Our results also demonstrated that staph phage K and vancomycin could synergistically reduce viable bacteria in both static and dynamic biofilms, even at subtherapeutic vancomycin levels (i.e., 33.75 µg/mL). However, phages alone were not able to reduce metabolically active bacterial cells present in the biofilms formed on the mesh.

The phage-antibiotic synergy in our experiments supports the results published by Kumaran et al. that showed synergy between staph phage SATA-8505 and cell-wall synthesis inhibiting antibiotics, including vancomycin, against static biofilms [45]. Dickey et al. examined a broad range of antibiotics and a PYO phage against *S. aureus* biofilms and found both antagonistic and synergistic combinations. They found that phage and antibiotic combinations worked best at lower antibiotic concentrations [51]. Most phage and antibiotic treatment studies, including those described above, have been conducted using planktonic cells or static biofilms [32]. Certain pathogenic strains of *S. aureus* form biofilms more readily under dynamic conditions due to an increased supply of oxygen [61] and genetic variations in the SCC*mec* gene [62]. Despite the demonstrated effect that growth conditions can have, only a small number of groups have used non-static biofilms when examining the effects of phage and antibiotic combination treatments. One of these studies used continuous-culture biofilms to show that a combination of gentamicin and phage SA5 resulted in synergistic reduction of *S. aureus*. Promisingly, no phage-resistance was detected after phage-antibiotic treatments [63].

Rising *S. aureus* resistance to vancomycin and other common antibiotics continue to render current antibiotic therapies ineffective [64,65]. To slow the spread of antibiotic resistance and explore alternative therapies, more research must be done to discover robust treatments against *S. aureus* biofilms. In the future, we plan to test a wider array of antibiotics and phages to determine how dynamic growth conditions impact different treatments. Some studies have shown that the addition of phages and antibiotics has the potential to treat biofilm infections in vivo [66,67], but more research is needed to better understand the mechanisms underlying phage-antibiotic interactions.

In support of prior research on biofilm treatments, our findings serve as a basis for phage and antibiotic combination treatments against *S. aureus* biofilm infections. The results presented here suggest that groups should consider the effect of biofilm-surface material, growth conditions, and antibiotic-phage identities when designing in vitro experiments. Although more research needs to be conducted, our study paints a promising future in the fight against biofilm infections on implantable devices.

## Figures and Tables

**Figure 1 viruses-15-00460-f001:**
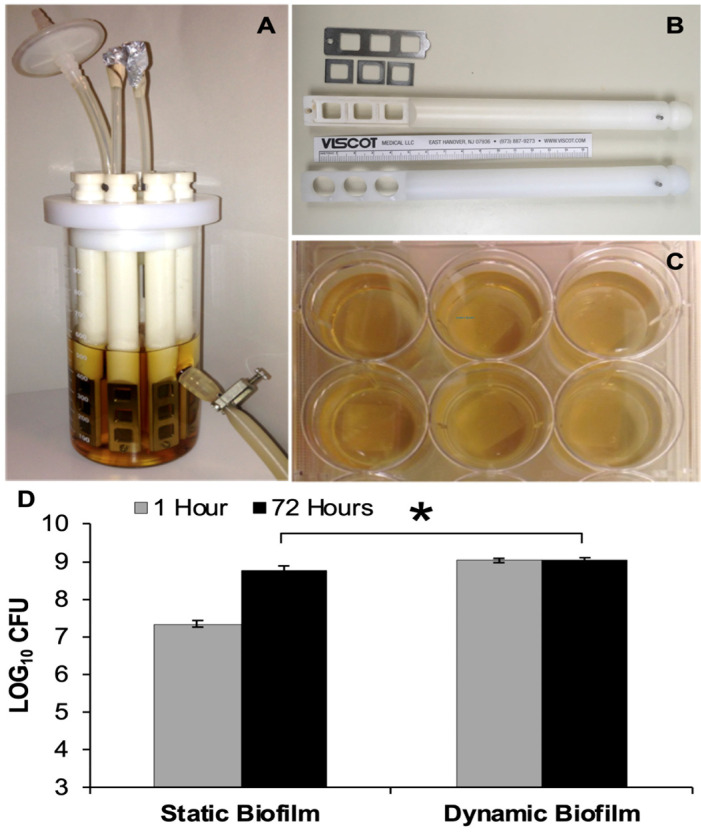
Biofilms Models. (**A**) CDC Biofilm reactor used to grow 48-h dynamic biofilms; (**B**) ‘In-house’ rod designed to hold three square PEEK membranes with metal frames versus manufacturer’s rod used to hold three PEEK disc coupons; (**C**) Multi-well plates used to grow 48-h static biofilms on PEEK membranes; (**D**) Bacterial load present on static and dynamic biofilms, which were transferred to small Petri plates and enumerated at t = 1 h and t = 72 h. Error bars represent the standard error of the mean. * Statistical significance (*p* ≤ 0.05).

**Figure 2 viruses-15-00460-f002:**
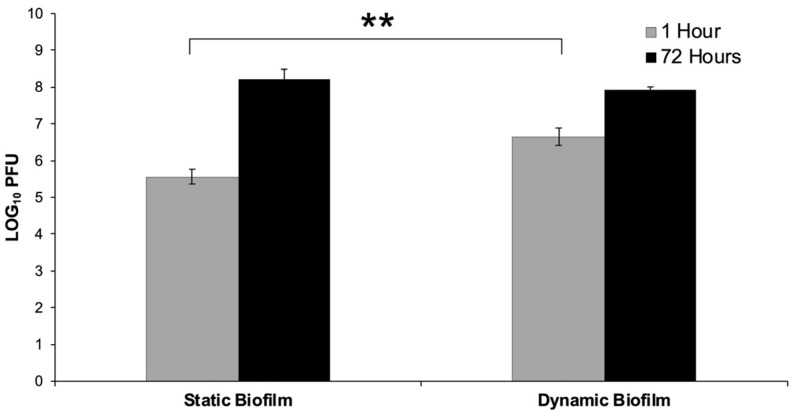
Phage Activity on Dynamic and Static Biofilms. 48-h old biofilms were treated with staph phage K 10^9^ PFU/mL. The phage titer present in the biofilms were enumerated at t = 1 h and t = 72 h. The average phage titer was 1.08 log units greater in the dynamic biofilms compared to the static biofilms at t = 1 h (** Statistical significance *p* ≤ 0.001). Error bars represent the standard error of the mean.

**Figure 3 viruses-15-00460-f003:**
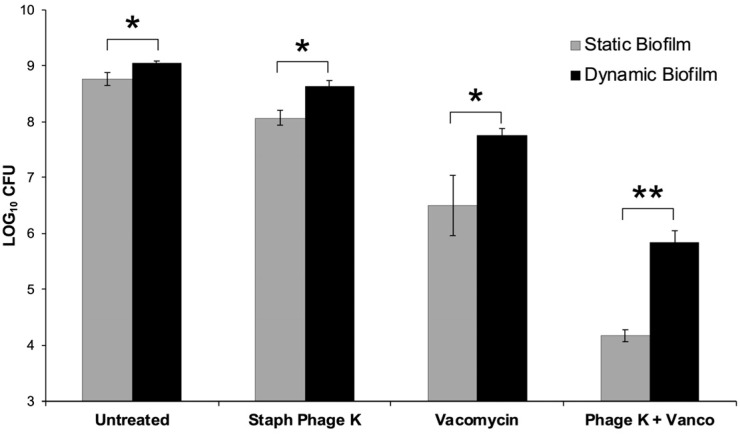
Comparison of treatment effects on dynamic versus static biofilms. 48-h old biofilms were treated for 72-h with: Staph phage K (10^9^ PFU); vancomycin (42 µg/mL); or a combination of phage and vancomycin. An untreated (control) group was set up for each biofilm type. There were statistically significant differences in bacterial loads between the static and dynamic biofilms post-treatment. Error bars represent the standard error of the mean. * Statistical significance (*p* ≤ 0.05). ** Statistical significance (*p* ≤ 0.001).

**Figure 4 viruses-15-00460-f004:**
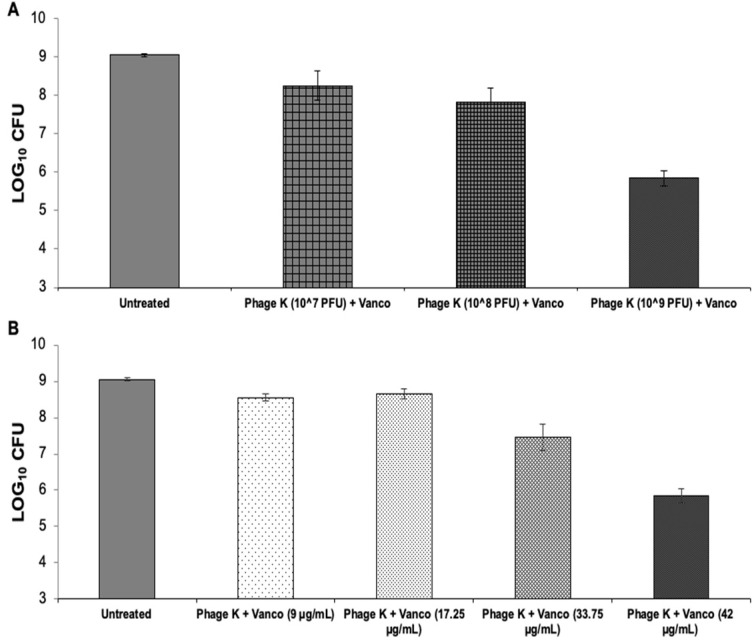
Vancomycin and staph phage K combinations—bacterial loads 72-h post-treatment on 48-h dynamic biofilms. (**A**) Biofilms treated with a combination of vancomycin at 42 µg/mL and staph phage K at the following concentrations: 10^7^ PFU/mL; 10^8^ PFU/mL; and 10^9^ PFU/mL. (**B**) Biofilms treated with a combination of staph phage K at 10^9^ PFU/mL and vancomycin at the following concentrations: 9 µg/mL, 17.25 µg/mL, 33.75 µg/mL and 42 µg/mL. Error bars represent the standard error of the mean.

**Figure 5 viruses-15-00460-f005:**
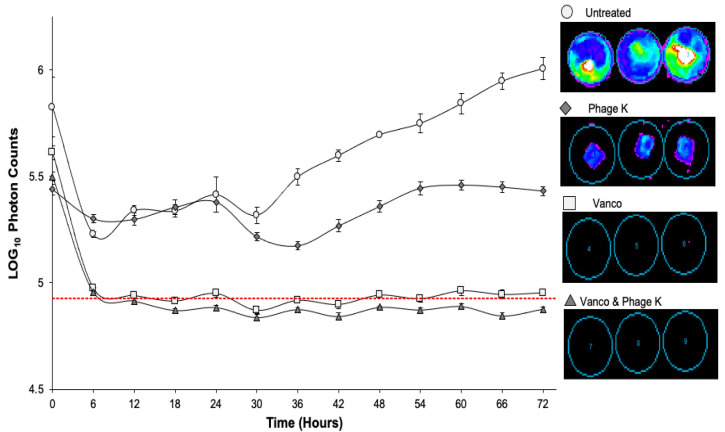
Bioluminescent activity of dynamic biofilms over 72-h treatment period. Vertical bars represent standard error of the mean. Dotted red line shows limit of bioluminescence detection of imaging system. Treatments included: untreated; vancomycin; staph phage K; combination of vancomycin and staph phage K. Luminescent intensity is displayed on a rainbow color scale with violet being least intense and red being most intense. Minimum and maximum intensities are shown to be 300 and 3000, respectively.

**Figure 6 viruses-15-00460-f006:**
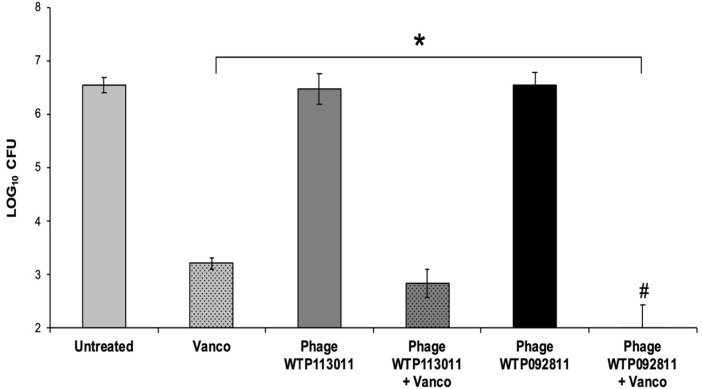
Vancomycin and staph phage WTP113011 and WTP092811 treatment of PEEK disc coupons. Bacterial loads 72-h post-treatment on 48-h dynamic biofilms grown on PEEK disc coupons. The discs were treated with phages WTP113011 and WTP092811 (10^8^ PFU/mL), vancomycin (40 µg/mL), or a combination of the two phages and vancomycin. (Error bars represent the standard error of the mean. * Statistical significance (*p* ≤ 0.05). # Average bacterial load < 2 CFU per biofilm.

## Data Availability

The data supporting reported results is available in the Appendix A accompanying this article.

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
