# Peer review of "Phage and Antibiotic Combinations Reduce Staphylococcus aureus in Static and Dynamic Biofilms Grown on an Implant Material"

_viruses, 2023, doi:10.3390/v15020460_

Round 1

Reviewer 1 Report

There are studies on the effect of antibiotics and bacteriophages on biofilms (see, for example, https://doi.org/10.3390/antibiotics8030103 or https://doi.org/10.1371/journal.pone.0209390, many of them). This manuscript considers the possibility of using a combination of vancomycin and bacteriophages against biofilms that can form on a material that has great potential for use in implants. It is the use of such material in this study that makes the manuscript unique. PEEK is hydrophobic; probably, it will be easier to form biofilms on it. This fact is an important reason for doing such research and writing this manuscript.

The manuscript has flaws:

1. line 106 has "...used on the MSSA membrane biofilms...". What is MSSA? Maybe this was meant by MRSA?

2. Figure 2 describes the activity of a bacteriophage. Is it necessary to standardize the number of bacteriophage particles? For example, something like the ratio of the number of phage particles to the number of bacterial cells. The reviewer thinks it would be more correct.

3. In the caption to figure 6 "... discs

were treated with WTP113011 and WTP092811, vancomycin, or a combination of the phages and vancomycin...", but the horizontal axis labels are completely different, Phage A here, Phage B, etc. This is misleading.

4. It seems that in the discussion section, the review of literature data (lines 338-371) is given more attention than the discussion of the results. The reviewer believes that this fragment of the discussion is redundant or its place is in the introduction section. But this point is debatable.

Author Response

  1. Experiments were conducted on both MSSA (methicillin-sensitive S. aureus) and MRSA (methicillin-resistant S. aureus). Staph phage K did not produce sufficient lytic activity on the MRSA strain we tested (i.e., SAP231), which is why we only presented experiments between Staph phage K and MSSA biofilms. Hence, we did not change “MSSA” to “MRSA” as commented by the reviewer because the information is correct.
  2. Figure 2 shows the concentration of phage particles detected/enumerated after they were added to the biofilms in order to determine if there was an increase in phage particles (i.e., phage amplification/activity) over a 72-hour period. We provided information about the ratios (i.e., MOIs) of phage to bacterial cells within the text of the Results section - we believe that it is more appropriate to show the concentration phage present at the start and end of the experiment rather than the MOI because it is easier for readers who don’t have a phage biology background to understand what is happening to the phage population. We don't agree that it is necessary to standardize the number of phage particles in Fig. 2.
  3. We have clarified our labeling for Fig. 6 and used the actual phage names instead of Phage A and Phage B – this was a version control issue that was missed by the corresponding authors prior to submitting our manuscript.
  4. The Discussion Section has been revised and some of the background information (i.e., review of literature) moved up to the Introduction section. We removed some of the redundant information included in the Discussion Section.

Reviewer 2 Report

In the MS entitled “Phage and antibiotic combinations reduce Staphylococcus aureus in static and dynamic biofilms grown on an implant material” Dr Joo and co-authors report the effect of usage of bacteriophages to overcome problems related to resistance of Staphylococcus aureus to antibiotics.  Such resistance represents the increasing impediment in modern treatment of diseases induced by the bacterium. The authors address specifically the infectivity related to biofilms on Polyether Ether Ketone (PEEK), a material used to manufacture orthopaedic implants. S. aureus was used as the biofilm-forming organism. The biofilms for the PEEK membrane experiments were obtained in either static or dynamic growth conditions and then treated with the antibiotic vancomycin, bacteriophages, and the combination of vancomycin and phages (the Staph phage K, WTP113011 and WTP092811).  Results obtained by the authors indicated that the most effective results were obtained using the combined treatment of phages wih the antibiotic. The most effective was the combination of vancomycin with the Staph phage K. The WTP113011 phage was not effective, while the activity of the WTP092811phage was noticeable only in the presence of vancomicin.

This is very interesting paper and well written; the methods are well described.

There are a few comments on this manuscript:

1.    While the MS is well written it would be more important if the discussion will be concentrated on the results obtained, It would be recommended to move parts of the discussion to the introduction: general overview (lines 338-347) and historical part related to the phages (lines 357-372 and lines 384-392),

2.    It is interesting that the authors have found (lines 394-403) that phages had “higher titre from dynamic biofilms than static biofilms even though the initial treatment titre was equivalent”. I am not sure that the dynamic films have “unique” architecture, possibly it should be said “distinct architecture”, which apparently is rather porous allowing better accessibility for phages to bacteria, while static biofilms make continue and less penetrable layers.

3.    The format of the MS in not consistent, the authors have to check formatting.

4.    Unfortunately, the authors did not provide any information on characterization of the phages used in their experiments. Are they dsDNA phages? Is it known to which family they belong Mioviridae, Siphoviridae, or Podoviridae?  Were the authors sure that the host for the WTP phages was S. aureus? It seems that these phages do not have specific recognition of this host and did not infect it. Apparently the WTP113011 phage belong to another group of phages compared to the Staph K and WTP092811 phages. The last one was a bit more active in the presence of vancomycin.

5.    It would be good to know how the authors assessed statistical significance of differences between phages in real numbers, which should be given in the text. It was not easy to find this information, taking in account that all data were normalised to the value of ten.

6.    There is a typo in the line 266, the text is not consistent with the legend for Figure 2.

7.    Line 299 and 304. Possibly it would be better to use the word “distinctive” in the expression “unique reduction”.

8.    It is rather difficult to compare the information which partly given in the text (lines 301-302) and partly in the figure 4 legend “Biofilms treated with a combination of Staph phage K at 109 PFU/mL and vancomycin at the following concentrations: 9 μg/mL; 17.25 μg/mL; 33.75 μg/mL; and 42 μg/mL. Error bars represent the standard error of the mean.“ These two bits of info should be close to each other.

9.    The formatting of lines 313-315 is a bit strange.

10. Why “Dotted red line shows limit of detection of imaging system” in figure 5? It is located above measurements obtained for Vanco and Vanco with PhageK. It seems that the authors were able to do reliable measurements in spite of low sensitivity of the device.

11. Line 373: “and burst size of MFP phages”. It is rather confusing, what do the authors mean: sizes of the phages or their amount after reaction? Why the comparison done with phage T4?

12. Line 383. The authors claim that “They found that phage and antibiotic combinations worked best at lower antibiotic concentrations [54]” Why is it important? How is it related to the current experiments reported in this MS?

Do the authors confirm this information?

Author Response

  1. Historical information previously found in the Discussion Section, which was not specifically associated with our results, has been moved to the Introduction section or been deleted if it was found to be redundant.
  2. The term ‘unique architecture’ has been changed to ‘distinct architecture’.
  3. The directions outlined in the ‘Instructions for Authors’ was followed during the development of this manuscript along with using the journal’s Word template. Please note that the instructions state that “we do not have strict formatting requirements”. However, we have standardized the spacing between paragraphs and the arrangement of the figures and legends so their layout in the manuscript is more uniform.
  4. We did not characterize the two new phages but isolated them using the MRSA strain as host. We have added the relevant information about using the single plaque purification method – which is now mentioned in the relevant Materials and Methods Section for clarification.
  5. Section 2.7 describes the statistical analysis used. We are now also including Supplemental Material with the calculated data for easy of verifying statistical analysis. No changes to the manuscript were made associated with this comment from Reviewer 2.
  6. Text connected to Figure 2 has been corrected to state 't=1 'rather than 't=0'.
  7. The word ‘unique’ has been changed to ‘distinct’.
  8. Figure 4 is now on the same page as the relevant text.
  9. Formatting issues have been standardized.
  10. Dotted red line represents the limit of bioluminescence detection. The word ‘bioluminescence' has been added to the legend to clarify the purpose of the dotted line.
  11. The term ‘burst size of MFP phages’ was used by Comeau et al. to indicate the number of progeny phage released and plaque size – not the size of the phages. We presented the Comeau et al. study to show the PAS (phage antibiotic synergy) effect, which we also saw from our experiments when combining phage and antibiotics. We have added more information on the PAS effect to clarify the original information in our manuscript.
  12. We have added information about using subtherapeutic levels of vancomycin, which when combined with phage could still result in a reduction in bacteria in a biofilm. Note that this information was not added immediately after the sentence about lower antibiotic concentrations (previous on line 383), but we hope this clarifies why using phage and antibiotic combination therapy can be beneficial. Our experiments do not confirm that lower antibiotic concentrations worked best, which is why we did not connect our results to that sentence but rather we cited the paper to show what others have seen positive effects when combining phage and antibiotics.

Reviewer 3 Report

Sound, interesting, and timely piece of work that addresses the important issue of bacterial drug resistance by evaluating the effectiveness of antibiotic/phage combinations to treat dynamically or statically generated Staphylococcus aureus biofilms. Although the scope of the study is rather limited (e.g. only one antibiotic is included in the tests), the interesting data obtained from combinations with different bacteriophages is likely to provide additional impetus towards using similar therapeutic alternatives for the treatment of insidious implant-related infections at the clinical level.

Nevertheless, the manuscript could be improved by taking care of several minor issues, mainly related to its format:

[1] Misspelled/inappropriate  words (e.g. "combing" instead of "combining" in line 12 of the Abstract) should be corrected.

[2] In all Figures indicating statistical significance for the data (Figs. # 1, 2, 3 and 6), the font size used for the asterisks is far too small, making them quite difficult to see in the printed version.

[3] In Figure #5, the fluorescence-showing image on the right side panel should be either presented in a larger size or deleted altogether. In its current size it fails to contribute any support for the soundness of the time-related fluorescence changes shown on the left side of the figure.   

Author Response

  1. ‘Combing’ has been corrected to ‘combining’ in the Abstract Section.
  2. The font size for the asterisks has been increased for all relevant figures to allow readers to see the symbol much clearer.
  3. The fluorescence images in Figure 5 have been enlarge to allow for better visualization. In addition, a sentence has been added to the text of the Results Section describing the observations seen with the bioluminescence images, as well as within the Discussion Section.